# Antibacterial Activities and Life Cycle Stages of *Asparagopsis armata*: Implications of the Metabolome and Microbiome

**DOI:** 10.3390/md21060363

**Published:** 2023-06-17

**Authors:** Christelle Parchemin, Delphine Raviglione, Anouar Mejait, Pierre Sasal, Elisabeth Faliex, Camille Clerissi, Nathalie Tapissier-Bontemps

**Affiliations:** 1Centre de Recherches Insulaires et Observatoire de l’Environnement (CRIOBE), Ecole Pratique des Hautes Etudes (EPHE), Université PSL, UPVD, CNRS, UAR 3278, 52 Av. Paul Alduy, CEDEX, 66860 Perpignan, France; christelle.parchemin@univ-perp.fr (C.P.); delphine.raviglione@univ-perp.fr (D.R.); anouar.mejait@univ-perp.fr (A.M.); sasal@univ-perp.fr (P.S.); camille.clerissi@ephe.sorbonne.fr (C.C.); 2Centre de Formation et de Recherche sur les Environnements Méditerranéens (CEFREM), UMR 5110 UPVD-CNRS, Université de Perpignan-Via Domitia, 52 Av. Paul Alduy, CEDEX, 66860 Perpignan, France; faliex@univ-perp.fr

**Keywords:** red algae, bacterial diversity, halogenated secondary metabolites, metabolomics, metabarcoding, multi-omics

## Abstract

The red alga *Asparagopsis armata* is a species with a haplodiplophasic life cycle alternating between morphologically distinct stages. The species is known for its various biological activities linked to the production of halogenated compounds, which are described as having several roles for the algae such as the control of epiphytic bacterial communities. Several studies have reported differences in targeted halogenated compounds (using gas chromatography–mass spectrometry analysis (GC-MS)) and antibacterial activities between the tetrasporophyte and the gametophyte stages. To enlarge this picture, we analysed the metabolome (using liquid chromatography–mass spectrometry (LC-MS)), the antibacterial activity and the bacterial communities associated with several stages of the life cycle of *A. armata*: gametophytes, tetrasporophytes and female gametophytes with developed cystocarps. Our results revealed that the relative abundance of several halogenated molecules including dibromoacetic acid and some more halogenated molecules fluctuated depending on the different stages of the algae. The antibacterial activity of the tetrasporophyte extract was significantly higher than that of the extracts of the other two stages. Several highly halogenated compounds, which discriminate algal stages, were identified as candidate molecules responsible for the observed variation in antibacterial activity. The tetrasporophyte also harboured a significantly higher specific bacterial diversity, which is associated with a different bacterial community composition than the other two stages. This study provides elements that could help in understanding the processes that take place throughout the life cycle of *A. armata* with different potential energy investments between the development of reproductive elements, the production of halogenated molecules and the dynamics of bacterial communities.

## 1. Introduction

Unlike terrestrial plants, marine algae are known for their ability to produce halogenated compounds linked to the particular availability of bromine, iodine and chlorine in marine environments [1,2]. Particularly, a high number of halogenated compounds have been isolated from red algae that harbour biosynthetic pathways for their production [1]. This production could be related to various ecological roles including chemical defences against the grazing or regulation of bacterial communities (also named microbiota) [1,2,3]. For example, halogenated furanones produced by *Delisea pulchra* [4,5], bromoform and dibromoacetic acid produced by *Asparagopsis armata* [6], polyhalogenated 2-heptanoone produced by *Bonnemaisonia hamifera* [7] and bromophycollides produced by *Callophycus serratus* [8] were reported as factors constraining bacterial communities.

Amongst the red algae known to produce halogenated compounds, those of the genus *Asparagopsis* have been extensively studied, leading to the identification of a wide variety of these molecules, mainly using gas chromatography–mass spectrometry (GC-MS) [6,9,10,11,12,13] and, more recently, liquid chromatography–mass spectrometry (LC-MS) [14,15]. Both species of this genus (*A. armata* and *A. taxiformis*) have a heteromorphic haplodiplophasic life cycle with different alternating stages: the plumose male or female gametophyte stage; the microscopic carposporophyte stage that develops on the female gametophyte and is protected inside gametophyte tissues called cystocarps; and the filamentous tetrasporophyte stage. Based on results from GC-MS analyses, several studies have reported differences in the composition of halogenated molecules between the tetrasporophyte and the gametophyte stages of *A. armata* [6,16,17]. Thus, Paul et al. [6] demonstrated that the levels of bromoform, dibromoacetic acid and bromochloroacetic acid were higher in the tetrasporophytes than in the gametophytes. Another study also reported that female gametophytes exhibited a higher abundance of bromoform than male gametophytes, particularly in the cell wall of the cystocarps [17]. These authors suggested an ecological role of these compounds via their potential to prevent grazing by herbivores. Indeed, it has been shown that the sea slug *Aplysia parvula* avoided consuming the female gametophytes of *A. armata* and, particularly, the cystocarps containing carsposporophytes [17]. It has also been shown that halogenated compounds produced by *A. armata* were involved in the control of epiphytic bacterial densities at the surface of this algae [6]. Several molecules isolated from this algae, such as bromoform and the recently described mahorones, have been reported as having antibacterial activities [6,18]. To our knowledge, only two studies concerning the antibacterial activity of several algae have included different stages of the *A. armata* life cycle. Both studies demonstrated high antibacterial activity for both stages and a significantly higher antibacterial activity for the extract obtained from the gametophyte than for the tetrasporophyte [19,20]. However, the authors did not carry out any analyses of the chemical composition. These studies clearly suggested possible interactions between bacterial communities and the halogenated molecules produced by the algae. In other words, we can hypothesise that the variation in the composition of halogenated molecules between the different stages of the algae could induce a variation in the composition of the bacterial communities. To our knowledge, only one study has focused on the differences in bacterial assemblages between different stages of the same algal species. It was conducted on the gametophytes and sporophytes of several brown algal species of the genus *Mastocarpus* and highlighted distinct bacterial communities between the two stages [21]. Morphology and chemical composition were two factors put forward by the authors to explain the observed variation, although no chemical investigations have been conducted to confirm this hypothesis [21]. Regarding the characterisation of bacterial communities associated with *A. armata,* few data are available to date. For example, Aires et al. [22] described bacterial communities associated with *A. armata* and *A. taxiformis* using metabarcoding and found that the two species harboured distinct bacterial communities, although they were collected at the same place. This study also highlighted functional genes associated with bacteria and those particularly linked to the production of secondary metabolites but not necessarily halogenated molecules, which could emphasize the role of microorganisms in the defence process of these algae [22]. Another study reported the identification of bacteria associated with the gametophyte stage of *A. armata* and their antitumor and antibacterial activities [23].

In this context, the main objective of our study was to explore the composition of the metabolome and bacterial communities between the different stages of *A. armata* in order to provide elements that could help in understanding the processes that take place throughout the life cycle of the algae. We used a metabolomics approach and a biphasic extraction focusing on apolar extracts that we previously reported as exhibiting antibacterial activity and containing a high diversity of halogenated molecules [24]. We also evaluated the variation of the antibacterial activity of algae extracts and used a bioguided fractionation combined with the calculation of the Pearson correlation to identify bioactive compounds. Then, we studied bacterial communities associated with the different stages of the life cycle of this algal species. Finally, we used multi-block statistical tools to evaluate correlations between metabolome and microbiota in order to highlight potential roles of the metabolites produced by the algae for the control of bacterial communities or, conversely, the role of bacteria in the production of bioactive metabolites.

## 2. Results

### 2.1. Variation of the Metabolome of the Three Life Stages

Metabolome variations were analysed via a metabolomics study on the apolar extracts of five replicates of each of the three following life stages of *A. armata*: the tretrasporophyte (T), the gametophyte (G) and the gametophyte with developed cystocarpes (GC) collected in Banyuls-sur-Mer where the species is invasive. After data processing and filtering, the matrix obtained from the LC-ESI—HRMS analysis included 261 features. A PCA was performed to evaluate the global variation of algal metabolome. The two first principal components (PC) explained 56.82% of the total variation. The scores plot showed a separation of GC from the two other stages along PC1, while T and G were separated along PC2 (Figure 1A). Each stage harboured a different metabolomic fingerprint (*p* = 0.012 for each pair). The hierarchical clustering analysis (Appendix A) of the matrix showed a similar clustering to that observed on the PCA score plot with two clusters, with one containing G and T samples and the other one containing GC samples.

We then performed a supervised statistical analysis using a PLS-DA model to determine which features mostly discriminated the three phases. The classification error rate (CER) of the model was low (CER = 0.01, *p* = 0.001), indicating a predictive model.

Variables with a high importance in projection, called VIP, with a score > 1.25 (77 variables) were selected, and, after redundancy elimination (fragments and adducts), a total of 32 features were considered. A molecular formula was proposed for most of the features based on exact mass, isotopic patterns, MS/MS fragmentation and Sirius annotation (Appendix A). The exploitation of the isotopic patterns allowed us to determine that most of the molecules were halogenated (23 over 32, 72%). Visualisation using a heatmap of the clustering of samples according to the relative abundance of these 32 molecules allowed the identification of three groups of discriminating molecules, most of which were common to G and T and more abundant than in GC (Figure 2A). For example, a molecule with the molecular formula C_5_H_2_Br_6_O_2_ (compound **1**, Figure 3), which also displayed the highest ion intensity on chromatograms (Appendix A), and another molecule that could be an analogue with the replacement of a bromine ion with a chlorine ion (C_5_H_2_Br_5_ClO_2_, compound **2**, Figure 3) were more abundant in G and T than in GC (Figure 2B). The identified molecules that were more abundant in GC were mainly low halogenated molecules containing bromines such as a dibromo acid derivative, bromochloroacetic acid and dibromoacetic acid (compound **3**, Figure 3) (Figure 2C). Finally, some features (i.e., M179T626, C_15_H_22_Br_6_O_2_ and C_15_H_23_Br_5_O_2_) were mostly detected in T (Figure 2A). Among them, both M179T626 (Figure 2D) and M451T626 (Appendix A) were barely detected in G and GC, corresponding to in-source fragments of pentabromopropen-2-yl tribromoacetate (C_5_Br_8_O_2_, compound **4**, Figure 3), which were previously isolated by our research team [25].

### 2.2. Variation in the Antibacterial Activity of the Three Life Stages

The antibacterial activity of the apolar extracts (0.5 mg/discs) of the samples of the three algal stages (5 biological replicates) was evaluated against 6 fish pathogenic bacteria: *Edwardsiella anguillarum* (Ea) (Gram−), *Lactococcus gervieae* (Lg) (Gram+)*, Tenacibaculum maritimum* (Tm) (Gram−)*, Vibrio anguillarum* (Va) (Gram−)*, V. harveyi* (Vh) (Gram−) and *Yersina ruckeri* (Yr) (Gram−). The extracts displayed antibacterial activity against all 6 bacteria (Table 1). The antibacterial activity of the three stages was significantly different for all bacteria, except for the two *Vibrio* spp. (Table 1). In addition, GC activity tended to be lower than that of the other stages, especially against Ea, Tm and Yr (Table 1). Against Lg, the measured inhibition diameters were much smaller than against the other bacterial species. Only T extracts showed activity against this bacterium, while G extracts showed very low and variable activity, and GC extracts showed no activity (Table 1).

A PCA was performed to evaluate the global variation of algal antibacterial activity (Figure 1B). The two first PCs explained 89.94% of the total variation. The scores plot showed a similar separation to that observed for the algal metabolome (Figure 1A) with the separation of the gametophyte with the developed cystocarps (GC) from the two other stages along PC1 and PC2 (Figure 1B). The covariation of the antibacterial activity and the metabolome was studied with a Mantel test and showed a positive correlation (r = 0.41, *p* = 0.025).

### 2.3. Identification of Candidate Molecules Responsible for Antibacterial Activity

The identification of candidate molecules responsible for the antibacterial activities was performed by bioguided (with *E. anguillarum*) fractionation on 100 g (dry weight) of powder of gametophyte of *A. armata*. To facilitate the extraction of large quantities of algae, extractions were realised by three successive macerations in MeOH:DCM (1:1). Like the apolar extract, this crude extract showed antibacterial activity against *E. anguillarum*, and the measured bacterial growth inhibition zone was 2.3 ± 0.1 cm. A bioguided fractionation was then performed on this extract using reverse-phase chromatography. Fractions eluted with 40% and 30% H_2_O-MeOH, from the first fractionation, exhibited 98.9 ± 0.4% and 97.6 ± 0.8% of bacterial growth inhibition, respectively (Appendix A), evaluated on 96-well plates (0.5 mg/mL). The LC-HRMS analyses of all fractions revealed that the two most active fractions harboured several intense ions (Appendix A) with monoisotopic masses at *m*/*z* 214.8352 [M−H]^−^, corresponding to compound **3** (Figure 3), and 527.2532 [M−H]^−^ (C_23_H_44_O_11_S), 555.2844 [M−H]^−^ (C_25_H_48_O_11_S) and 566.5085 [M−H]^−^, corresponding to compound **1** (Figure 3). Peak integrations were manually performed, and the correlations between the determined peak areas and the bioactivity of the fractions were calculated for 14 major ions detected in active fractions (Appendix A). Finally, five ions with monoisotopic masses at an *m*/*z* of 444.6488 (C_5_H_3_Br_4_ClO_2_) (compound **5**, Figure 3), 334.7556 (C_4_H_3_Br_3_O_3_) (compound **6**, Figure 3) and 488.5982 (C_5_H_3_Br_5_O_2_) (compound **7**, Figure 3) and masses corresponding to compound **1** and **2** (Figure 3) exhibited a strong correlation (r > 0.8) with the bioactivity (Appendix A). These molecules were seemingly structure-related as, on MS/MS spectra (Appendix A), 1, 2, 6 and 7 displayed a fragment with a monoisotopic mass at *m*/*z* 248.7552 [M−H]^−^, corresponding to a [CBr_3_]^−^ ion fragment, and 2 and 5 displayed a fragment with a monoisotopic mass at *m*/*z* 204.8066 [M−H]^−^, corresponding to a [CBr_2_Cl]^−^ ion fragment. A comparison with recent published data [15] led us to assign structures for **1**, **2**, **5** and **7** as polyhalogenated 2,4-diones (Figure 3). For compound **6**, which was not observed by Thapa et al. [15], we proposed the most probable structure based on fatty acid biosynthetic pathway coherence and fragmentation patterns (Appendix A).

Active fractions were further fractionated by semi-preparative HPLC leading to 14 fractions of which two, F9 and F10, induced 99 ± 0.0% and 93.7 ± 0.3% of bacterial growth inhibition, respectively (Appendix A). These fractions still presented the same ions (Appendix A) that were highly correlated with the bioactivity (Appendix A). However, while most of the ions presented a similar range of intensity between the first and second fractionation, a loss of intensity (Appendix A) was observed for the one corresponding to compound **1**. At this stage, and despite a subsequent fractionation by semi-preparative HPLC, we did not achieve the purification of any of the halogenated compounds present in the active fraction.

### 2.4. Analysis of Bacterial Communities Associated with A. armata Stages

The variations of the bacterial communities associated with the same five replicates for each of the three algal stages for which metabolome and antibacterial activity were described in Section 2.1 and Section 2.2 were then assessed. The metabarcoding sequencing of the 16S rRNA gene resulted in a dataset of 2,235,756 reads. After filtering and rarefying to the lowest number of sequences (24,710) (Appendix A; Appendix A), the final dataset was finally composed of 3075 amplicon sequence variants (ASV) (Appendix A). The α-diversity indices were calculated. No significant differences were observed between the three stages for evenness (Table 2). In contrast, T displayed significantly higher values than the other algal stages for Chao1 and Shannon indices (Table 2). 

Differences in β-diversity were assessed using Bray–Curtis dissimilarities. The PCoA showed a separation of bacterial communities in accordance with the different stages (Figure 4A). The average distance to median (GC = 0.1855, G = 0.2476, T = 0.2611) indicated no significant difference in the intragroup (intra-stage) bacterial composition (*p* = 0.39). The T samples were separated from the other two stages according to PCo1 (Figure 4A) and showed a significantly different bacterial composition from G and GC (*p*_(GC_T)_ = 0.015 and *p*_(G_T)_ = 0.015), which did not separate according to PCo2 and the result of the pairwise permanova test (*p*_(GC_C)_ = 0.189). By taking a closer examination of the structure of bacterial communities, we established that the sequences belonged to 283 genera corresponding to 156 families, 87 orders, 35 classes and 21 phyla. The α-proteobacteria and γ-proteobacteria (Pseudomonadota) and Bacteroidia (Bacteroidota) dominated the bacterial communities in all algal stages at the class level (Figure 4B). At the family taxonomic rank (Appendix A, Appendix A), Saprospiraceae (Bacteroidia) dominated in a similar range in all algal stages, representing 26% on average of the ASVs (GC: 7517 sequences, G: 6402 sequences and T: 5080 sequences out of 24,710 in total). Abundances in other families were more variable among the different algal stages, with Flavobacteriaceae (Flavobacteriia) representing 11% on average (2637 sequences out of 24,710 in total) of the total sequences of T and only 2% on average (GC: 531 sequences, G: 549 sequences out of 24,710 in total) in the two other algal stages. In a similar way, “Other” families (representing families with an abundance <0.02%, i.e., five sequences or less) accounted for 31% on average (7761 sequences out of 24,710 in total) of the total sequence of T and less than 15% on average (GC: 2603 sequences, G: 3355 sequences out of 24,710 in total) of that of the other two algal stages. In contrast, other families such as Thiotrichaceae were more represented in GC and C than in T samples with 14%, 11% and 5% on average of the ASVs for GC (3350 sequences), G (2778 sequences) and T (1313 sequences), respectively (Appendix A).

By calculating the indicator value index, we revealed that the majority of the discriminant ASVs (311 belonging to 39 distinct families) were significantly associated with T, while only 40 ASVs and 63 ASVs were significantly associated with G and GC, respectively (Appendix A). Furthermore, the most abundant discriminating ASVs (representing > 1% of the total sequences) (Appendix A; Figure 5) were almost equally associated with the three algal stages, with eight ASVs associated with GC, six with G and five with T (Appendix A; Figure 5). The clustering of samples on the heatmap representation of these most abundant discriminating ASVs showed a clustering similar to the projection of samples of each algal stage with the PCoA (Figure 4A). GC and G samples clustered together, and T samples were in a second group and harboured very specific bacterial ASV (Figure 5). Three Cellvibrionaceae with an unidentified genus (ASV_13, ASV_27 and and ASV_38) were in very low abundance in T (0.06% on average) compared to their abundances in G and GC (0.9% and 2% on average). In a similar way, the *Thiotrix* and *Cocleiomonas* genera (ASV_37 and ASV_44), both belonging to the Thiotrichaceae family, were more abundant in G and GC (0.9% and 1% on average) than in T (0.03%) samples. Conversely, a *Roseobacter* genus (Rhodobacteraceae) (ASV_83) and a *Lewinella* genus (Saprospiraceae) (ASV_18) were not sequenced at all in C and GC samples (Figure 5, Appendix A) but represented an average of 1% and 6% of the total sequences in T samples, respectively. Finally, two genera each of Flavobacteriaceae, *Winogradskyella* and *Croceitalea* were also significantly associated with T samples.

The potential functional profile of bacterial communities associated with the algae was explored using the predictive metagenomics tool Tax4Fun2. It allowed the prediction of a total of 6625 functions (Appendix A). The indicator value index was used and 433, 1201 and 4306 predicted functions were associated with GC, G and T, respectively. According to the KEGG Mapper reconstruct tool, they were mostly associated with global and overview maps (mainly metabolic pathways; biosynthesis of secondary metabolites; and bacterial metabolism in diverse environments), carbohydrate metabolism, and amino acid metabolism (Appendix A). We then focused on the most abundant predicted functions (abundance > 0.1%) (108 GC, 107G and 107T) of which 28 were associated with GC, 72 were associated with G and 24 were associated with T (IndVal) (Appendix A). None of the predicted functions associated with the bacterial diversity of GC were involved in secondary metabolites but rather in metabolic pathways and membrane transport. In contrast, the functional diversity associated with the bacterial community of both G and T was mostly involved in metabolic pathways and the biosynthesis of secondary metabolites (Appendix A). Some specific predicted functions with a lower abundance (<0.1%) associated with the bacterial diversity of T were involved in the breakdown of macroalgal polysaccharides (beta-agarase, alpha-agarase, kappa-carrageenase, lambda-carrageenase, iota-carrageenase and oligo-alginate lyase). Finally, the predicted functions involved in dehalogenase production were associated with the bacterial diversity of G and GC (2-haloacid dehalogenase for GC; haloacetate dehalogenase for G).

### 2.5. Correlation between Metabolite and Bacterial Compositions in Algal Stages

The correlation between metabolite and bacterial compositions of the five replicates of each of the algal stages was assessed by a Mantel test and a multiblock PLS-DA (DIABLO). The Mantel test showed no significant covariation between both datasets (r = 0.16, *p* > 0.05). In contrast, the scores plot (Appendix A) and correlation scores (0.93 on dimension 1 and 0.89 on dimension 2; Appendix A) of the multiblock PLS-DA analysis between the datasets showed good congruence and covariation between the bacterial community and the metabolite compositions for each algal stage. The three algal stages were well discriminated with the built model (CER = 0.06, *p* = 0.001). The first dimension discriminated T from the other groups, while the second component discriminated GC from G (Appendix A). A clustered image map was built to observe the clustering of the 60 metabolites and 32 ASVs selected by the model. Three clusters were distinguishable, with (1) 41 variables discriminating G, (2) 32 discriminating T and, finally, (3) 19 discriminating GC (Appendix A). The cluster of the variables discriminating G was mostly composed of ASVs (29) (Appendix A), while the clusters of the variables discriminating T and GC were mostly composed of 28 and 14 metabolites, respectively.

The three clusters can also be observed on the correlation network, which was built with the most discriminant variables that were highly positively or negatively correlated (threshold = 0.7) (Figure 6, Appendix A).

Notably, three clusters contained halogenated (including compounds **1**, **2**, **3**, **4** and **7**) and non-halogenated metabolites. Regarding the ASVs present in the correlation network, with the exception of ASV_38 (Cellvibrionaceae_NA), ASV_13 (Cellvibrionaceae_NA) and ASV_44 (Thiotrichaceae_Cocleimonas), all other ASVs that were correlated with metabolites (halogenated or not) had a relatively low abundance (<1%) in the algal samples (Appendix A). The cluster of the variables discriminating T was mainly composed of positively correlated variables, whereas the correlations were more mixed for the cluster of variables discriminating G and GC. Some halogenated molecules including compounds **1**, **2** and **4** were positively correlated with the four ASVs discriminating T, all of which were Flavobacteriales (ASV_182, ASV_369, ASV_780 and ASV_1372). They were also strongly negatively correlated with some ASVs. We can distinguish compounds **1** and **2** but also **7**, that were negatively correlated with ASVs discriminating GC, such as a Campbellbacteria variant (ASV_1320), a Saprospiraceae variant (ASV_169) and a 0000069-P22 variant (ASV_363) and those, including compound **4**, negatively correlated with ASVs discriminating G, such as three Cellvibrionaceae variants (ASV_13, ASV_38 and ASV_51) (Figure 6).

## 3. Discussion

*Asparagopsis armata* and *A. taxiformis* are two red algae known for their production of halogenated molecules [26,27,28,29]. Several studies reported differences in the concentration of some targeted halogenated molecules between the different life stages of the genus [6,16,17]. In this study, we first aimed at exploring the metabolome and antibacterial activity variation of the different life stages of *A. armata*. Secondly, we performed a bioguided fractionation in order to identify the molecule(s) responsible for the antibacterial activity. Thirdly, we studied the variation of microbiota composition between the three algal stages, and finally, we assessed the complex interactions between the chemistry and the microbiota datasets.

### 3.1. The Three Stages of the Life Cycle of A. armata Harboured Distinct Metabolome Compositions

Using a metabolomics approach, we discovered that the three collected stages harboured different metabolome compositions. By studying the most discriminating molecules and their isotopic patterns, we were able to highlight that they were mainly halogenated molecules (25 over 33). Among them, haloacetic acids such as dibromoacetic acid, bromochloroacetic acid and dibromomethyl succinic acid were more abundant in GC (gametophyte with developed cystocarps) than in the other two stages, the contents of these molecules not being significantly different between G and T. These results differ from those of Paul et al. [6] who quantified targeted halogenated molecules contents in *A. armata* and found consistently higher levels of dibromoacetic acid and bromochloroacetic acid in T compared to G. These differences may suggest that the location and the collection period could have an impact on the results found. In GC, the abundance of the three above-cited molecules could be linked to the development of the cystocarps on the female gametophyte. Indeed, Vergès et al. [17] suggested that the accumulation of not only bromoform but also dibromoacetic acid, both of which were reported to be correlated, in the cystocarps walls of *A. armata* [6] may be related to the need to protect the fragile parts of this alga, such as its reproductive elements [17]. In their study, the cystocarp walls were the least consumed parts of the algae in preference tests performed on the sea slug *Aplysia parvula* [17]. A monitoring of the metabolome during the gradual transition from the G to the GC stage could reveal the increased abundance of these molecules and help to better understand their roles.

Then, we discovered that highly halogenated molecules, such as compounds **1** (C_5_H_2_Br_6_O_2_) and **2** (C_5_H_2_Br_5_ClO_2_), were significantly more abundant in G and T than in GC. The structures of molecules **1** and **2** were established by comparing the data obtained by Thapa et al. [15] who proposed their data on the basis of MS^n^ fragmentation patterns and suggested that they may be precursors of smaller halogenated molecules, such as dibromomethane and chlorodibromomethane, by hydrolytic cleavage. We could hypothesise that, in G and T, these molecules could be part of the storing strategy of halogenated resources, while in GC, these molecules could have undergone the transformations mentioned above, leading to a high abundance of small deterrent halogenated molecules. Thapa et al. [15] also reported the presence of not only vanadate-dependent haloperoxidases in *A. taxiformis*, which could be involved in the production of bromoform by the algae [15,30], but also other halogenated compounds as these enzymes are known to catalyse the biosynthesis of halogenated molecules [2,31,32]. Then, we discovered that compound **4** (C_5_Br_8_O_2_) was only detected in T. In a previous study, we have proposed this perbrominated compound as a potential chemotaxonomic marker of the gametophyte stage of *Asparagopsis* spp. as it was only detected in the gametophyte samples of *A. taxiformis* [24]. It was thus interesting to observe the presence of this molecule in the tetrasporophyte stage of this species. A differential expression of haloperoxidases, which could then lead to the differential production of halogenated molecules, could explain the observed differences in the abundance of not only compound **4** but also other halogenated compounds. The preferential expression of a gene encoding for bromoperoxidase was already reported in the sporophyte stage of the red algae *Pyropia yezoensis* [33]. The authors showed that it was actively expressed in filamentous sporophytes, leading to bromoform production, but repressed in leafy gametophytes under normal growth conditions, which is in accordance with our hypothesis.

### 3.2. Highly Halogenated Molecules May Be Responsible for Some Antibacterial Activity

By the screening of the antibacterial activity of the apolar extracts of the stages of the alga against six pathogenic bacteria of marine organisms, we found significant differences between the targeted stages. These results are in line with those of Salvador et al. [20] who studied the antibacterial activity against human pathogens of several Iberian algae, including the tetrasporophyte and the gametophyte stages of *A. armata,* and demonstrated significantly different antibacterial activity between the two stages [20]. However, in contrast to our study, a higher antibacterial activity was found for the gametophyte than for the tetrasporophyte. Similarly, the activity of the extracts from different stages of 26 cultivable algae species, including the gametophyte and tetrasporophyte stages of *A. armata*, was studied against pathogenic bacteria in aquaculture [19]. In this study, the dichloromethane extracts of both stages of *A. armata* were among the most active of algal extracts. The authors also measured higher inhibition diameters for the gametophyte than for the tetrasporophyte stage [19]. The differences in activity observed between these studies and our study could be related to the composition of the extracts, which depends on the extraction method used, and the extraction method differed between all the studies. Indeed, we used a biphasic extraction, whereas the results mentioned above were obtained with aqueous extracts [20] and dichloromethane extracts [19]. The antibacterial activity of the GC extracts was significantly lower than that of the T extracts against three of the bacteria. Moreover, the extracts of GC and G were not active against Lg. In general, the inhibition diameters measured against this bacterium were much lower than those measured against the other bacteria. We could hypothesise that this may be related to the fact that Lg is a Gram-positive bacterium. However, in the literature, several studies on the antibacterial activity of *A. armata*, which included both Gram-positive and Gram-negative bacteria, did not report major differences or a higher activity of the algae when measuring activity against Gram-positive compared to Gram-negative bacteria [6,20]. Thus, the observed lower activity may be more species-related, with Lg likely being less sensitive to our algae extracts than those of other bacteria.

The metabolomics analyses allowed us to discover that highly halogenated molecules were in higher abundance in G and T than in GC. The antibacterial activity of highly halogenated molecules, isolated from *A. taxiformis*, was already described. This is the case for compound **4**, which has already been reported for its antibacterial activity against two strains of bacteria belonging to Flavobacteriaceae [25] and pentabrominated mahorones [18]. In addition, via bioguided fractionation and the calculation of the Pearson correlation, we identified structurally related halogenated compounds (**1**, **2** and **5**–**7**) as being potential responsible for the observed antibacterial activity. Compounds **1** and **2** also appeared as VIPs for the discrimination of algal stages. Unfortunately, and despite the precautions taken, like Thapa et al. [15], who have already tried to isolate compound **1** to confirm its structure but did not succeed because of its lability, we failed to complete the purification of the active molecules. Regarding compound **1**, we also observed a decrease in the corresponding chromatographic peak during the successive fractionation steps, which suggests the instability of this compound in our purification conditions. Therefore, we could not confirm the activity of the molecules cited above; however, with the correlations measured and the activities already described for other molecules, we could consider that highly halogenated molecules may be involved in some of the variations in the antibacterial activity observed in vitro. Their ecological roles remain to be determined. The localisation of these molecules in the algal tissues by MALDI imaging mass spectrometry and the analysis of *Asparagopsis* exometabolome by LC-MS may provide some answers by indicating whether these molecules are stored in the specialized cells already observed in *A. armata* [6,34] and released on the algal surface and the immediate surrounding environment.

### 3.3. Microbiota Composition Differs throughout the Three Stages of the Life Cycle of A. armata

We then studied the composition of the microbiota of the same samples of the three algal stages as in Section 3.1 using metabarcoding analysis. Pseudomonadota (α-proteobacteria and γ-proteobacteria) and Bacteroidota (Bacteroidia) were the dominant phyla (and classes) of the bacterial communities, which is consistent with what is commonly found in macroalgae [3]. Similarly, α-proteobacteria (Rhodobacterales and Rhizobiales) were also found to be dominant in the tetrasporophyte of *A. taxiformis* in previous studies [35]. Saprospiraceae (Bacteroidia) was the most represented family (26%) in our samples. This was in accordance with previous studies that reported that this family was very abundant in samples of *Asparagopsis* spp. and could be a constitutive part of the microbiota of the genus *Asparagopsis* [22].

The comparison of bacterial communities between the different stages of *A. armata* clearly showed that the alpha diversity (evenness, Chao1 and Shannon indices) of the T samples was significantly higher than that of the GC and G samples, for which there was no significant difference. This specific bacterial diversity of T was associated with a greater number of predicted functions, which were calculated using Tax4Fun 2 [36], than for the GC and G stages. These observations are in line with those of another study that explored the microbiota of the gametophyte and sporophyte of *Mastocarpus* spp. and found that the specific diversity of the bacterial community of the two stages of the alga was different. However, unlike our study, the sporophyte samples of *Mastocarpus* spp. presented a lower Chao1 index and, thus, a lower specific diversity than the gametophyte samples [21]. The genetic host identity [37], the microbial communities analysed (surface microbial communities compared to whole microbial communities in our study), and the collection site (Queen Charlotte Basin in British Columbia compared to the Mediterranean Sea in our study) could be some explanations for the observed differences between both studies. Indeed, it is well known that the environment and the location of the samples have an influence on microbial composition [38].

The molecules produced by the algae could also impact the composition of the bacterial community [3]. In our study, we found no correspondence between the clustering analyses of the molecular composition and the bacterial community composition of the different algal stages nor any covariation between the molecular composition and bacterial community composition (Mantel test). These results suggest that there is no relationship between chemical composition and bacterial community composition for any algal stage. Nevertheless, there may be correlations between specific metabolites and specific bacteria as seen in our study and further discussed below. We can also hypothesise that other molecules, not observed in our study, could be involved in bacterial colonization. As an example, polysaccharides that constitute the cell walls of macroalgae are sources of nutrients that some bacteria can break down [3], and their composition may differ between gametophytes and tetrasporophytes [39]. In our study, among the predicted functions of the identified bacterial communities, some, significantly associated with T, were involved in the degradation of polysaccharides for which T composition differed from that of the gametophyte [40]. However, the most abundant bacteria associated with the tetrasporophyte were Flavobacteriaceae, which are known for their ability to degrade algal polysaccharides [41,42,43]. Apart from the predicted functions involved in the degradation of polysaccharides, some of the most abundant predicted functions (>0.1%) associated with the bacterial community diversity of G and T were involved in the biosynthesis of secondary metabolites, which was not the case for GC. This could be an additional explanation for the differences in antibacterial activity observed between G, T and GC. However, the functions were predicted using Tax4Fun2, and further metabolic details would be revealed by metagenomics combined with metatranscriptomics [22].

Apart from chemistry, morphology may also influence bacterial composition [21,44]. The gametophyte and the tetrasporophyte of *A. armata* have very distinct morphologies to the point that they were long considered to belong to different genera until the life cycle of the algae was unravelled [28]. The tetrasporophyte has a pompom-like shape and measures 2 to 3 cm maximum, whereas the gametophyte, bearing numerous branches, can measure up to 40 cm [45]. Due to these different morphologies, the two algal stages interact differently with their surrounding environment and with water flow, which also influences biofilm formation and potentially provides a diversity of habitats for bacterial communities [46,47].

Among the most abundant and discriminant ASVs associated with tetrasporophytes, Flavobacteriaceae were well represented, notably with the genera *Croceitalea* and *Winogradskyella*. The above-mentioned bacterial genera were also encountered in the older basal parts of the brown algae *Taonia atomaria*, which, compared to medial and apical parts, also harboured the highest bacterial density and diversity [48]. The author hypothesised that the basal part of the algae may be subjected to a longer period of exposure to bacterial communities and thus may be exposed to a high succession of communities, which could explain the higher bacterial density and diversity. Given that, in Banyuls-sur-Mer, the tetrasporophyte is persistent throughout the year, whereas the gametophyte is only present from late January to June, these features could explain the great bacterial diversity of *A. armata* tetrasporophyte.

### 3.4. The Overall Analyses Performed Suggest Complex Interactions between the Host and Its Microbiota

Although the unsupervised analysis suggested no significant relationship between chemical and bacterial community compositions, the supervised multi-omics analysis allowed us to identify discriminant variables that correlated with each other in both datasets. The discriminant ASVs identified in the model were correlated with both halogenated and non-halogenated molecules. These ASVs were mostly present in a low abundance in the algal samples, meaning that more than just the most abundant bacteria are potentially involved in chemical interactions with the host.

In this study, among ASVs negatively correlated with compounds **1**, **2** and **7** and with other halogenated molecules, three were abundant in GC: a Saprospiraceae variant (Bacteroidota), a Campbellbacteria (Patescibacteria) variant and a 0000069-P22 variant (Patescibacteria). Compound **4** was negatively correlated with three Cellvibrionaceae (Pseudomonadota) variants that were abundant in G. The genus of these ASVs is unknown, and, as the genome of these bacteria has not yet been sequenced, we lack information to hypothesise their potential functional roles. However, dehalogenase functions have been predicted and associated with GC (2-haloacid dehalogenase) and G (haloacetate dehalogenase). This is of particular interest as compound **4** is a haloacetate and was negatively correlated with ASVs that were abundant in G. Thus, in our study, bacteria that are negatively correlated with halogenated molecules could produce specific enzymes that could perform dehalogenation and recycle halogenated molecules, which would lead to a decrease in the abundance of halogenated molecules and could explain negative correlations. In the literature, these types of enzymes were also found in the Flavobacteria *Zobellia galactanivorans* (Bacteroidota) [49], in a Rhodobacteraceae (Pseudomonadota) [50] and in *Psychromonas ingrahamii* (Pseudomonadota) [51]. Negative correlations may also be explained by a potential susceptibility of the above-cited genera to halogenated molecules, which were therefore less sequenced in the life stages producing these molecules.

It was interesting to observe the positive correlations of compound **4** with Flavobacteriaceae as, in a previous study performed by our research team, the antibacterial activity of this molecule against the species of this family was reported [25]. We could have, therefore, observed negative correlations instead. This could be explained by differences in susceptibility at the species level. The general observation of the high abundance of this family in samples of the tetrasporophyte is interesting as some genera of this family were reported as potential causative agents for algal disease partly because of their ability to degrade polysaccharides [52,53,54]. However, the Flavobacteriaceae family is composed of a high number of genera, which are probably not all pathogenic for the algae. Apart from the two Flavobacteriaceae, two other members of the Flavobacteriales were also present in the T cluster and positively correlated with halogenated molecules. It is not excluded that members of this bacterial order could be involved in the biosynthesis of halogenated molecules by the algae, which could explain the positive correlations. However, the interpretation of the observed correlations is risky and must be fastidious because they could be related to different causes as mentioned. The cultivation of the identified bacteria of interest could be used to test the different hypotheses but is not always possible. The use of metagenomics combined with metatranscriptomics could also help by providing data on the link between bacterial community structures and functions in these communities [22,55].

Finally, the main results and hypotheses are summarized in Table 3.

## 4. Materials and Methods

### 4.1. Chemicals

For sample preparation, methyl-tert-butyl-ether (MTBE) of HPLC grade was purchased from Honeywell Riedel de Haen™ (Seelze, Germany), and methanol (MeOH) of HPLC grade, dichloromethane (DCM) and water of HPLC grade were purchased from VWR™ (Fontenay-sous-Bois, Paris, France). Water used for fractionation was ultra-pure water (18.2 MΩ.m, total organic carbon <10 ppb and <10 CFU/mL) obtained by purifying water with a PURELAB Chorus 1 system (ELGA LabWater, Lane End, High Wycombe, HP14, UK). For UHPLC-HRMS analysis, ultra-pure water was used, and acetonitrile LC-MS grade was purchased from Carlo Erba (Val de Reuil, Normandy, France). Formic acid 99% (for LC-MS analysis) was obtained from Carlo Erba (Val de Reuil, Normandy, France).

### 4.2. Biological Materials

For metabolomics, metabarcoding analyses and evaluation of antibacterial activity, five replicates for each of the three stages of the algal life cycle (Table 3 and Appendix A) were sampled in May 2022 in the natural marine reserve in Banyuls-sur-Mer, France (lat. 42.482230; long. 3.137175) where the species is invasive. These stages include samples of tetrasporophyte (T) and two sample types of gametophyte, without cystocarps (G) and with developed cystocarps (GC) (Table 3 and Appendix A). Physically contiguous algae were pooled and considered to be one single individual while individuals separated by at least one meter from others are considered as different individuals. Five individuals (biological replicates) for each algal life cycle were collected and processed separately. Immediately after collection, algae were cleaned of epiphytes. One part was stored at −80 °C for metabarcoding analysis, and the other part was freeze-dried and stored at −20 °C until subsequent analyses.

Due to a limited amount of algal material collected in 2022, the bioguided fractionation was performed on a pool of *A. armata* (gametophyte) samples collected between February and May 2021 in Banyuls-sur-Mer (lat. 42.482230; long. 3.137175) and processed as indicated above.

### 4.3. Algal Extraction

Freeze-dried algae were ground to obtain a homogeneous powder. For metabolomics, samples were extracted as described in [24]. A biphasic extraction method was used. Briefly, 50 mg of each dried and crushed algae sample (5 replicates for each algal stage) was extracted with a mixture of MeOH and MTBE and vortexed. Then, H_2_O was added, and the whole solution was vortexed. Extraction was performed in an ultrasonic bath, and the final mixture was centrifuged to obtain two phases. Equal volume of apolar phase was collected for each sample. Solvent was evaporated with a centrifugal vacuum evaporator GENEVAC EZ-2^TM^ (SP Scientific, Warminster, PA, USA), and the dried samples were stored at −20 °C until LCMS analyses. Quality control (QC) samples were prepared by pooling equal volume from all extracts. For the antibacterial activity assays only 3 replicates for each algal stages were extracted using the same biphasic extraction but using 250 mg of algae in order to obtain sufficient material for the assays.

Finally, for bioguided fractionation and to facilitate the extraction of larger quantities of algae, *A. armata* powder (100 g dry weight) was extracted by three successive macerations in MeOH:DCM (1:1).

### 4.4. Metabolomics

#### 4.4.1. Chemical Analyses

Extracts were analysed following the protocol described in [24]. Apolar extracts were solubilised in 1 mL MeOH. LC-HRMS analyses were performed with a Vanquish UHPLC system from ThermoScientific (Waltham, MA, USA) equipped with a Q Exactive™ Plus mass spectrometer with an electrospray ionization source. Metabolite separation was performed on a C18 UHPLC column (Luna^®^ Omega 1.6µm Polar C18 100 A LC Column 100 × 2.1 mm, Phenomenex, CA, USA). The mobile phase consisted of a mixture of H_2_O + 0.1% formic acid (solvent A) and acetonitrile + 0.1% formic acid (solvent B). The flow rate was 400 µL/min. The program was set up at 2% B for 1 min, followed by a linear gradient up to 100% B for 10 min and then maintained for 5 min in isocratic mode. The analysis was followed by a return to initial conditions for column equilibration for 4 min for a total runtime of 20 min. Extracts were randomly injected, alternating the quality control sample injections every 5 samples.

The mass spectrometer analyser parameters were set as follows: sheath gas flow rate, auxiliary gas and sweep gas flow rate set to 45 arbitrary units (a.u.), 15 a.u. and 2 a.u., respectively; capillary and gas temperature were set to 320 °C and 250 °C, respectively; the S-lens RF level and spray voltage were set to 60 V and 3.20 kV, respectively. MS/MS acquisition consisted of one full scan mass spectrum and 5 data-dependent MS/MS scans. Parameters for the full MS experiments were set as follows: the resolution was 70,000, the automatic gain control was 3E6 ions, the maximum injection time (IT) was 100 ms, and a scan mass window of 100–1500 *m*/*z* was used. For the dd-MS2/dd-SIM experiments, the resolution was 17,500, AGC target was 1e5 ions, and maximum IT was 50 ms. For each MS/MS scan, the top 5 most intense ions, taking into account an isolation window of 4.0 *m*/*z* and a fixed first mass of 50.0 *m*/*z*, were fragmented. Finally, normalized collision energy^TM^ (NCE) ranged between 25, 35 and 45 eV.

#### 4.4.2. Data Treatment

Data acquisitions were performed using Xcalibur 4.1.31.9 (Thermo Fisher Scientific, Waltham, MA, USA). Raw data were converted to mzML files with MSconvert (version 3.0, from Proteowizard library, Palo Alto, CA, USA). mzML files were uploaded and processed using the Galaxy web platform (version 3.3) [56,57]. The workflow used for data pre-processing and used parameters are published on the Galaxy Workflow4Metabolomics platform at https://workflow4metabolomics.usegalaxy.fr/u/christelle_parchemin/w/workflowparcheminalgae (publication date: 22 October 2021) and are available in Appendix A. Briefly, the pre-processing consisted of chromatographic peak detection (Galaxy Version 3.12+galaxy0, accessed on 2 July 2022), followed by a peak grouping, a loess/non-linear “PeakGroups” retention time adjustment, a peak filling and “CAMERA” peak annotation. A matrix of features with peak intensity, *m*/*z* value and retention time was generated. A clean-up step was performed in order to eliminate all features that were significantly detected in blanks. Then, an “intra-batch” signal correction was applied using the “Batch correction” function with a “loess” regression model [58]. This step was followed by a second clean-up according to feature’s CV in pool QC injections [59]. Finally, redundancies due to isotopes were manually eliminated (only monoisotopic mass was kept). For identification, most probable molecular formula was determined using Sirius (v4.9.15 [60,61,62,63,64,65,66,67], accessed on 13 February 2023), characteristics isotopic clusters, MS/MS spectra and comparison with the literature.

#### 4.4.3. Multivariate and Statistical Analyses

Statistical analyses were performed using RStudio environment v2022.02.3 (R v4.2.0) and MetaboAnalyst 5.0 [68]. Autoscaling was applied on data. Principal component analysis (PCA) (prcomp {stats}) was used to evaluate the global dispersion of algae metabolome. Permutation tests (Adonis 2 {VEGAN} and pairwise.perm.manova {RVAideMemoire }) were used to test the discrimination between groups. PLS-DA (plsda {mixOmics} [69]) was used for detection of features contributing the most to the model, and putative identification of the features was performed as mentioned in “4.4.2 Data treatment”. Permutation test based on cross-model validation (MVA.test and pairwise.MVA.test {RVAideMemoire}) was applied to validate PLS-DA model and to test the significance of the discriminations according to defined factors. Kruskal–Wallis tests followed by Nemenyi post hoc tests were used ({Stats}{PMCMRplus}) to test the significance of the differences in variable importance in projection (VIP) feature intensity among groups. Dendograms and heatmaps (distance measure: Euclidean; clustering algorithm: ward) were drawn on MetaboAnalyst 5.0 [68] to evaluate the grouping of samples and variables.

### 4.5. Antibacterial Activity

Apolar phase antibacterial activity was tested on several bacterial species. The recent promising studies on the potential for aquaculture of *A. armata* and *A. taxiformis* [70,71,72,73,74,75] led us to select fish pathogenic bacteria for our tests: *Edwardsiella anguillarum* (DSMZ-27202, “Ea”, Gram−), *Lactococcus garvieae* (CIP102507T, “Lg”, Gram+), *Tenacibaculum maritimum* (CIP103528T, “Tm”, Gram−), *Vibrio anguillarum* (CIP63.36T, “Va”, Gram−), *V. harveyi* (CIP103192T, “Vh”, Gram−) and *Yersina ruckeri* (CIP82.80T, “Yr”, Gram−). They were maintained at −80 °C, respectively, in liquid medium (Appendix A) supplemented with 30% (*v*/*v*) glycerol. Prior to use, all bacteria were revived from glycerol stocks and grown for 24 h at 26 °C on agar plates (except *Lg*, which grows at 37 °C). An isolated colony was then cultivated in liquid medium for 12 h. Agar plates were flooded with fresh bacterial broth (DO_620_ nm = 0.1) and were set aside to dry. Assays of extracts were performed using the disc diffusion method [76]. Briefly, 10 µL of extract was placed onto a 0.6 cm sterile paper disc (final concentration: 0.1 mg/disc). Discs were placed under the hood to allow evaporation of the solvent. Following evaporation, the discs were placed onto the surface of the inoculated agar. The plates were incubated for 24 h at 26 °C or 37 °C. Discs soaked with antibiotics (Appendix A) were used as positive control, and MeOH was used as negative control. Areas of inhibited bacterial growth were observed as clear halos (zones) around the discs. The diameter of the zone of bacterial growth inhibition was measured, and each reported measurement including the size of the disc. Kruskal–Wallis tests followed by Nemenyi post hoc tests were used to test the significance of the differences in antibacterial activity among groups.

For faster testing of a larger number of samples, fractions from the bioguided fractionation (see Section 4.6) were assayed in 96-well microplates and were only tested against *E. anguillarum*. Fractions were first solubilised at a concentration of 20 mg/mL in 100% DMSO and then diluted in sterilised water to reach a concentration of 5 mg/mL. Next, 180 µL of fresh 12 hours’ bacterial culture (DO_620_ nm = 0.1) was deposited in the well, to which 20 µL of fraction was added (final concentration of fraction 0.5 mg/mL and 2.5% of DMSO). Bacterial culture, medium, and solvent (DMSO) were also deposited in wells to serve as controls. After 24 h of growth, optical density at 620 nm was measured. Percentage of growth inhibition was calculated as follows: (((OD_bacteria+solvent_ − OD_medium_) − (OD_bacteria+fraction_ − OD_medium_))/(OD_bacteria+solvent_ − OD_medium_)) × 100.

### 4.6. Bioguided Fractionation and Identification of a Candidate Molecule Responsible for the Antibacterial Activity

Bioguided fractionation was performed on 100 g (dry weight) samples of *A. armata* gametophyteto to obtain 17.6 g of crude extracts via three successive macerations of algal powder in MeOH:DCM (1:1). Then, each crude extract was fractionated by flash chromatography under vacuum on a C-18 pre-packed column (45 g) (Interchim, puriflash IR 60, C18 50 UM, Montluçon, France) using a step gradient of 100% H_2_O, 90%H_2_O-MeOH, 80% H_2_O-MeOH, 70% H_2_O-MeOH, 60% H_2_O-MeOH, 30% H_2_O-MeOH, 20% H_2_O-MeOH, 10% H_2_O-MeOH, 100MeOH and 100EtoAc to create 11 fractions. The resulting fractions were evaporated, weighed and re-suspended in MeOH at the concentration to be tested and assayed for antibacterial activity. The most active fractions were combined and further fractionated using an HPLC (Waters 1525) coupled to a UV detector (Waters 2487) set to 220 and 330 nm. Metabolite separations were performed on a reverse-phase column (Luna^®^ 5µ PFP (2) 100 A, LC Column 250 × 10 mm, Phenomenex, Torrance, CA, USA). The mobile phase consisted of a mixture of H_2_O + 0.1% formic acid (solvent A) and acetonitrile + 0.1% formic acid (solvent B). The flow rate was 2.5 mL/min. The program was set up at 40% B for 4 min, followed by a linear gradient up to 100% B for 21 min and then maintained for 5 min in isocratic mode. The analysis was followed by a return to initial conditions for column equilibration for 8 min for a total runtime of 40 min. The fractionation resulted in 14 fractions. All fractions were assayed and analysed by LC-HRMS/MS as described in 4.4.1—Chemical Analyses. The peaks of the most intense ions on the chromatograms were manually integrated, and the correlation between the area of each peak and the activity measured in all fractions was assessed using a Pearson correlation {stats}.

### 4.7. Metabarcoding

#### 4.7.1. DNA Extraction, PCR and Sequencing

DNA extractions were conducted on approximately 1 cm^2^ of each of the five replicates of the three algal stages using the ZymoBIOMICS DNA Miniprep Kit (ref. D4300, ZYMO RESEARCH, Torrance, CA, USA) according to the manufacturer’s protocol. The 16S rRNA gene of bacterial communities was amplified and sequenced using the variable V3V4 loops (341F: 5′-CCTACGGGNGGCWGCAG-3′; 805R: 5′-GACTACHVGGGTATCTAATCC-3′) [77]. Paired-end sequencing (300 bp read length) was performed at Dalhousie University (Integrated Microbiota Resource, Halifax, NS, Canada) on the MiSeq system (Illumina) using v3 chemistry according to the manufacturer’s protocol. Raw sequence data are available in the Sequence Read Archive database (BioProject ID PRJNA948342).

#### 4.7.2. Sequence Analyses

The DADA2 package [78] (truncLen = c(260,230); maxN = 0; maxEE = c(2,2); truncQ = 2) was used to define amplicon sequence variants (ASV) and computed taxonomic affiliations using Silva database (release 138, December 2019). The dataset was filtered for singletons. Rarefaction curves of species richness were computed using the {phyloseq} R package and the ggrare function. The rarefy_even_depth function was used to subsample datasets. The estimate_richness function was used to compute alpha diversity metrics (Chao1, evenness and Shannon). Lastly, the tax_glom function was used to obtain abundances at different taxonomic ranks (from genus to phylum).

#### 4.7.3. Multivariate and Statistical Analyses

We performed a non-parametric Kruskal–Wallis test (because normality of residuals was rejected (Shapiro test)) to compare alpha diversity metrics between life stages. We then computed pairwise comparisons between group levels (post hoc analyses) using the Nemenyi test (kwAllPairsNemenyiTest {PMCMR plus}). Principal coordinate analysis (hereafter named PCoA) (ordinate, {phyloseq}) was computed to describe compositions of amplicon sequence variants (ASV) between samples using Bray–Curtis dissimilarities. Average distance to median was studied with the betadisper function {VEGAN}. Permutational multivariate analysis of variance (hereafter named PERMANOVA) was used to compare bacterial composition between life stages using 999 permutations (adonis2, {vegan}). Then, pairwise comparisons between group levels were computed using pairwise PERMANOVA (pairwise.perm.manova{RVAideMemoire}). We used variant abundances with default parameters to predict functional profiles using Tax4Fun2 [36]. This analysis provided a table of relative KEGG ortholog (KO) abundances. The reconstruct tool from KEGG Mapper (available at https://www.genome.jp/kegg/mapper/reconstruct.html, accessed on 19 December 2022) was used to identify metabolite pathways associated with the predicted functions. We used an indicator value index (hereafter named IndVal) and 999 permutations (indval, {labdsv}) [79] to identify specific taxa and functions associated with the different life stages. Heatmaps of relative abundances of specific ASVs and predicted functions were computed using the function heatmap ({stats}). For all analyses, the threshold significance level was set at 0.05.

### 4.8. Multi-Omics

The correlation between the variation of the metabolome and the variation of the bacterial community composition of each of the five replicates of the three algal stages was calculated via a Mantel test using Spearman correlation and a permutation test (mantel and vegdist {Vegan}). Correlation between the two datasets was further explored using the multiblock model DIABLO from the MixOmics package [80]. DIABLO allows the creation of a model that maximizes the covariance between datasets in order to identify the most correlated and discriminating variables according to a defined factor. Here, the model allows the identification of most correlated ASVs and metabolites involved in the discrimination of each algal stage. Optimal number of components (2) was determined with the perf() function. Number of metabolites (60) and ASVs (32) were determined using the tune.block.splsda() function. Finally, the model was validated by a cross validation test (DIABLO.cv(), {RVAidememoire}).

## 5. Conclusions

To conclude, we have explored conjointly, for the first time, the variations of the metabolome composition, the antibacterial activity and the bacterial community composition of three stages of the life cycle of *A. armata*. We have shown that there were variations between the algal stages but no covariations between the metabolome and the bacterial community composition. Gametophyte harbouring developed cystocarps (GC) was associated with the overexpression of low halogenated molecules, lower bacterial diversity and lower antibacterial activity, in contrast to the tetrasporophyte stage (T). We have shown that an abundance of highly halogenated molecules was different between algal stages and could be involved for part of the variation of the antibacterial activity between the different algal stages. However, their ecological roles remain to be determined. The tetrasporophyte stage harboured a significantly higher specific bacterial diversity, which was associated with a greater number of and diversity between predicted functions than that of the GC and G samples. While the Saprospiraceae appeared to be a constitutive part of the microbiota of the three stages, Flavobacteriaceae were very specific to the tetrasporophyte, but their functions for this stage remain to be determined. Finally, this study provided cues that could help in understanding the processes that take place throughout the life cycle of *A. armata* by building a hypothesis regarding the different energetic investments between cystocarps development, the production of halogenated molecules and bacterial community dynamics.

## Figures and Tables

**Figure 1 marinedrugs-21-00363-f001:**
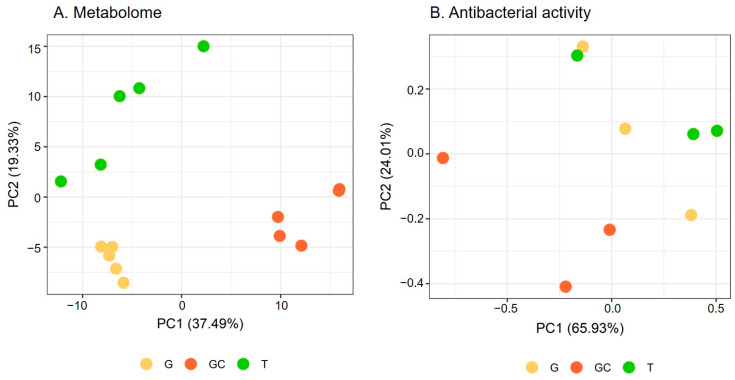
(**A**) Metabolome and (**B**) antibacterial activity scores plots of *A. armata* gametophyte “G”, with developed cystocarps “GC”, and tetrasporophyte “T” stages. (**B**) refers to results mentioned in Section 2.2.

**Figure 2 marinedrugs-21-00363-f002:**
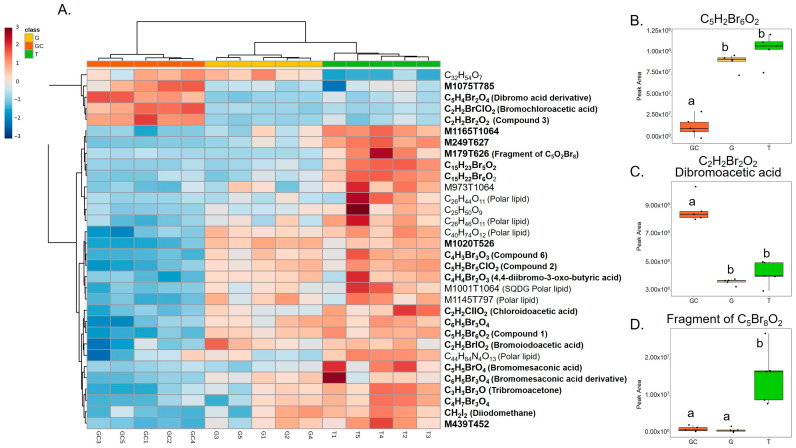
(**A**) Heatmap representation of top 33 VIP features for the discrimination of the three algal stages (the gametophyte, “G”; the gametophyte with cystocarps, “GC”; and the tetrasporophyte, “T”) detected with LC-HRMS and their putative identification. Clustering: ward. Distance: Euclidean. Putative identification in bold represents halogenated molecules. (**B**) On the right, box plots of peak area of C_5_H_2_Br_6_O_2_, (**C**) dibromoacetic acid and (**D**) C_5_Br_8_O_2_ in the three types of samples. Letters a and b represent distinct groups based on Nemenyi post hoc tests between algal stages (*p* < 0.05).

**Figure 3 marinedrugs-21-00363-f003:**
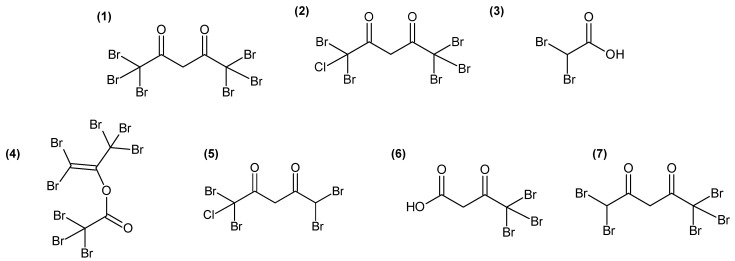
Chemical structures of molecules involved in the discrimination of the three algal stages (**1**–**4**) and of molecules with a correlation > 0.8 (Pearson coefficient) with the bioactivity of *A. armata* fractions (**1**,**2**,**5**–**7**).

**Figure 4 marinedrugs-21-00363-f004:**
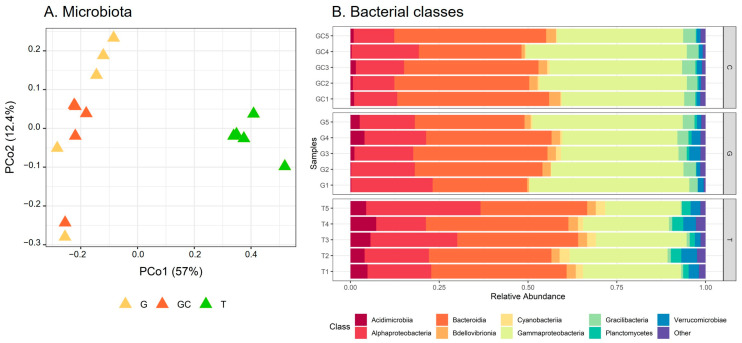
(**A**) Scores plot of microbiota of the three *A. armata* stages. (**B**) Relative abundance of bacterial classes associated with the three *A. armata* stages (the gametophyte, “G”; the gametophyte with cystocarps, “GC”; and the tetrasporophyte, “T”).

**Figure 5 marinedrugs-21-00363-f005:**
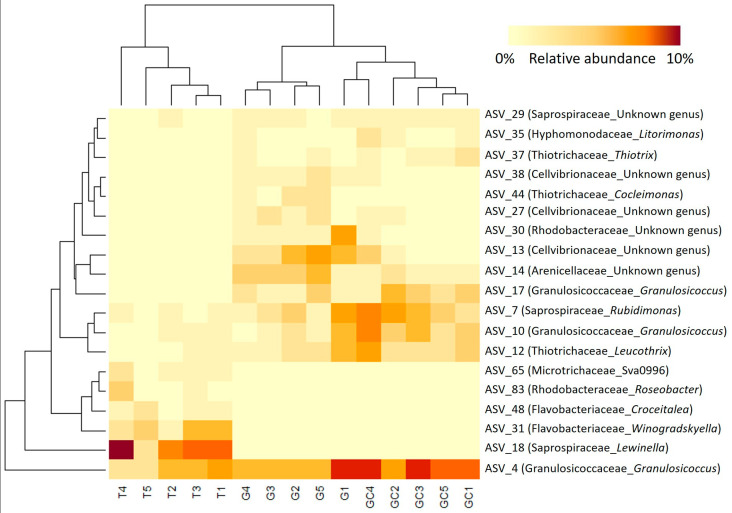
Heatmap of the most abundant (>1% of the total number of sequences) and discriminant ASVs of the three *A. armata* stages (the gametophyte, “G”; the gametophyte with cystocarps, “GC” and the tetrasporophyte, “T”). The lines represent the ASVs, the columns represent the samples, and the colours represent the relative abundance of each ASV in each sample.

**Figure 6 marinedrugs-21-00363-f006:**
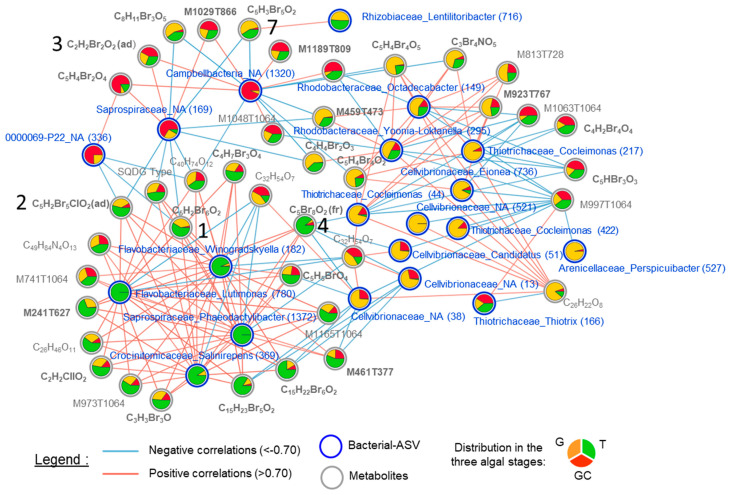
Correlation network of metabolites (grey circles and names) and ASVs (blue circles and names) selected by the DIABLO (multiblock sPLS-DA) analysis with a correlation threshold of 0.7. Negative correlations are represented by the blue lines, and positive correlations are represented by the red lines. Relative abundance in each algal stage (the gametophyte, “G”; the gametophyte with cystocarps, “GC” and the tetrasporophyte, “T”) is represented with the pie charts. Metabolite names in bold represent halogenated compounds. The numbers in brackets in Bacterial–ASV names represent the ASV number (see Appendix A). The numbers **1**, **2**, **3**, **4** and **7** refer to the metabolites in Figure 3.

**Table 1 marinedrugs-21-00363-t001:** Diameter of inhibition (including the size of the discs—0.6 cm) measured as indicator of the antibacterial activity of extracts (0.5 mg/discs) of the 3 types of algal stages (the gametophyte with cystocarps, “GC”; the gametophyte, “G”; and the tetrasporophyte, “T”) against 6 different bacterial species. Letters (a and b) represent distinct groups based on Nemenyi post hoc tests between algal stages for each bacterium (*p* < 0.05).

	GC	G	T	*p*
Ea	1.7 ± 0.2 ^a^	2 ± 0.06 ^a,b^	2.2 ± 0.1 ^b^	0.034
Lg *	0.6 ± 0	0.6 ± 0.06	0.8 ± 0	0.030
Tm	1.6 ± 0.1 ^a^	2 ± 0.06 ^b^	1.9 ± 0.1 ^a,b^	0.050
Va	2.0 ± 0.06	2.1 ± 0.06	2.0 ± 0.06	0.141
Vh	1.4 ± 0.4	1.2 ± 0.4	1.1 ± 0.4	0.610
Yr	1.4 ± 0.1 ^a^	1.6 ± 0.1 ^a,b^	1.7 ± 0.06 ^b^	0.042

***** For Lg, although a Kruskall–Wallis test indicated significant differences among groups, the groups were not significantly different with each other according to the Nemenyi post hoc tests.

**Table 2 marinedrugs-21-00363-t002:** α-diversity indices: Chao1, evenness and Shannon, of the three types of algal stages (the gametophyte with cystocarps, “GC”; the gametophyte, “G”; and the tetrasporophyte, “T”). Letters (a and b) represent distinct groups based on Nemenyi post hoc tests between algal phases (*p* < 0.05).

	GC	G	T	*p*
Evenness	0.78 ± 0.03	0.80 ± 0.04	0.84 ± 0.02	0.069
Chao1	513.4 ± 92.4 ^a^	632.5 ± 235.4 ^a,b^	920.7 ± 186.6 ^b^	0.031
Shannon	4.87 ± 0.3 ^a^	5.1 ± 0.5 ^a,b^	5.7 ± 0.3 ^b^	0.021

**Table 3 marinedrugs-21-00363-t003:** Summary of the main findings and hypotheses.

Algal Stages	Metabolome	Antibacterial Activity	Microbiota Diversity and Composition	Conclusions and Hypotheses
General observation	Significant differences between GC-G, GC-T and G-T	Significant differences between GC-T	Significant Differences between GC-T and G-T	Highly halogenated molecules may be responsible for a part of the antibacterial activity No major covariation between metabolome and microbiota (Mantel) Correlations between specific metabolites and ASVs
GC 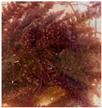	Abundance in C_2_H_2_Br_2_O_2,_ C_2_H_2_BrClO_2_ and C_5_H_4_Br_2_O_4_	Lower than T	Lower diversity than T Bacterial community dominated by the same ASVs as G	Distinct metabolome that may be related to the development and protection of reproductive elements
G 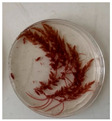	Composition similar to T with abundance in highly brominated molecules (C_5_H_2_Br_6_O_2_ and C_5_H_2_Br_5_ClO_2_)	Not significantly different from the two others (but closer to T)	Intermediate diversity between GC and T Bacterial community dominated by the same ASVs as GC	C_5_H_2_Br_6_O_2_ and C_5_H_2_Br_5_ClO_2_ may be responsible for a part of the antibacterial activities
T 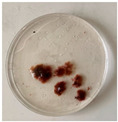	Abundance in C_5_Br_8_O_2_ barely detected in GC and G	Higher than GC but not significantly different from G	Diversity higher than GC and G Harbours some specific ASVs not shared with GC or G	Are morphology and annual persistence major factors influencing microbiota composition? Are differences in chemical composition involved in the bacterial diversity or vice versa? Are the greater differences in gene expression involved in specific biosynthetic pathways?

## Data Availability

Raw LC-ESI(-)-HRMS data were deposited and are publicly available in the MassIVE platform, number MSV000091636. Raw metbarcoding sequence data are available in the Sequence Read Archive database (BioProject ID PRJNA948342).

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
