# Peer review of "Antibacterial Activities and Life Cycle Stages of Asparagopsis armata: Implications of the Metabolome and Microbiome"

_marinedrugs, 2023, doi:10.3390/md21060363_

Round 1
Reviewer 1 Report

Just one minor comment
Reviewer 2 Report
The paper is well-written and sound. The only main concern is on the antimicrobial activities. 5 mg/mL is a very high concentration for an extract/fractions. More concentrations are absolutely necessary to properly prove the antimicrobial activities looking for a dose-response effect.
Author Response
R1 : The paper is well-written and sound. The only main concern is on the antimicrobial activities. 5 mg/mL is a very high concentration for an extract/fractions. More concentrations are absolutely necessary to properly prove the antimicrobial activities looking for a dose-response effect.
Authors would like to acknowledge the Reviewer for his constructive critics that helped enhancing the manuscript.
Regarding the concentration of extracts and fractions. The concentrations 10mg/mL and 5mg/mL are not the final concentrations for the assay. Indeed, the final concentration for the disc diffusion assay is 0.1mg/disc and for the assays in 96 well plates 0.5 mg/mL. These concentrations are lower than those that may have been used in antibacterial assays of these algae.
E.g. :
Pinteus et al. 2015 : Use of Asparagopsis armata dichloromethane extract 0.3 mg on 12 mm disc
Genovese et al. 2012 : Discs containing each 2 mg of the extract were placed onto the inoculated plates
And Marino et al. 2016 : “Each extract (100 mg, equivalent to 3 g of lyophilized algal sample) was dissolved in 1000 μl of absolute ethanol, and 20 μl were applied to sterile filter paper disks. After solvent evaporation, the disks (containing 2 mg of the extract) were placed onto the inoculated plates.
However, the final concentration was not indicated in the results part. To facilitate the understanding of the antibacterial activity part we have added it (L. 153-154).
Reviewer 3 Report
Manuscript: marinedrugs-2403978
The manuscript is comparing the metabolome, antimicrobial activity, and microbiome in the three stages of the macroalgae Asparagopsis armata: tetrasporiphyte, gematophyte and gematophyte with the female reproductive organ cystocap. It focusses also on the known halogenated compounds produced by the algae and their abundance in the different life stages.
Overall, the manuscript is well written and scientifically valuable. However, there are some issues that need to be addressed.
General comments:
Asparagopsis armata can be easily distinguished from A.taxiformis by its harpoons in the gematophyte phase, yet the tetrasporophyte phase can only be distinguished by the cell size (Nı´Chuala´in et al 2004). Thus, though the two species have overlapping growth area, it still makes sense that the authors may indeed sampled A.armata, However, as the paper is discussing metabolomics that may be derived from the sampling of tetrasporophyte, perhaps a molecular taxonomic analysis of this phase should be added to verify that the tetrasporophyte is indeed armata. Another concern is that the author declares that they have tested gametophyte samples from different years (May 2021 and Feb-May 2020), to compensate for the limited amounts. Thus, this needs to be carefully addressed and validated so that this is not affecting the data presented. Have the authors compared the metabolome of the samples from both years? Is there any "ground truth" – repeated patterns of compounds? Also, the authors attribute most of the activity to the halogenated compounds, yet there is evidence that many of the active compounds are not necessarily halogenated.
Specific comments:
- Line 3: the "?" in the title- is there a question there? It does not read as a question. Either remove or rephrase as a question.
- Line 125: VIP appears several times in the text only as an acronym. Please, please provide the full name for the first time in use.
- Line 143:“previously isolated by our research team [25]” The ref is not complete! Please provide the full ref. "Reverter, M.; Tapissier-Bontemps, N.; Banaigs, B.; Sasal, P.; Calvayrac, C.; Mazzitelli, J.-Y.; Tintillier, F. Composé antibactérien et antiparasitaire 2022."
- Line153-155: though it may be clear to microbiologists that this G means Gram negative or positive. to avoid confusion with the G of gematopyte, it is suggested that you will use a different label or even the full "Gram" at this point.
- Line 178- 179 “The identification of candidate molecules responsible for the antibacterial activities was performed by bioguided fractionation on 100g (dry weight) powder of gametophyte of A. armata.” Only for the gametophyte? In the methods it’s written that this has been done for all three life stages (4.6). And are these samples collected a year before and at a different time of the year than the other samples in this study. This needs to be clarified. It is ok that samples were isolated from the extract they were most abundant, but is this comparable to the activity and metabolome of the other samples?
- Line 182- “and the measured bacterial growth inhibition zone was of 2.3 ± 0.1 cm.” On which bacterial strain? And again, only gametophyte extract?
- Lines 217, 354, 449: A armata – remove underline and use italic.
- Line 302: “Correlation between metabolite and bacterial compositions in algal stages” Was this correlation done between the same samples origin? Or the bacterial compositions from the samples collected in 2021 and the metabolite of the samples collected a year before? This theme is repeated and needs to be clarified.
- Line 366-367-“Indeed, it was already reported high concentration of some targeted halogenated molecules, such as bromoform, in the cystocarp walls of A. armata which…” You reported that in your samples there was less antibacterial activity and fewer brominated molecules in the GC than in the T and G. How does this fact align with the previous results you discussed?
- “Then, we found out that compound 4 (C5Br8O2) was only detected in T. In a previous study, we have proposed this perbrominated compound as a potential chemotaxonomic marker of the gametophyte stage of Asparagopsis spp. as it was only detected in the gametophyte samples of A. taxiformis [24].” This is another example why the authors must provide molecular evidence that the tetrasporophyte they collected is indeed of armata and not mistaken as one.
- Line 420: “We could hypothesize that it may be related to the fact that Lg is a Gram-positive bacterium” that can also be attributed to the seasonality, extraction method or the microbiome and environmental conditions.
- Line 583- “For metabolomics and metabarcoding analyses and evaluation of antibacterial activity, three stages of the algal life cycle (Supplementary method 1)”. In the “Supplementary method 1” there are only photo of the different stages. I suggest referring to the photo in Table 3 instead and to remove it from the supplementary section.
References: please go over the references and make sure they are all searchable
Reviewer 4 Report
This was interesting work which was tackled in a methodical way with in depth analysis. You achieved what you set out to do but interestingly for me (and you alluded to this) you have uncovered many more questions which require to be answered.
Only one small thing: In your supplementary material you stated that the E. anguillarum was grown in Lubria broth, should that have read Luria broth.
Author Response
This was interesting work which was tackled in a methodical way with in depth analysis. You achieved what you set out to do but interestingly for me (and you alluded to this) you have uncovered many more questions which require to be answered.
-Authors would like to acknowledge the Reviewer for his reading and comment that helped enhancing the manuscript.
Only one small thing: In your supplementary material you stated that the E. anguillarum was grown in Lubria broth, should that have read Luria broth.
-This was indeed a mistake that we corrected, thank you for pointing it out.